The effects of island forest restoration on open habitat specialists: the endangered weevil Hadramphus spinipennis Broun and its host-plant Aciphylla dieffenbachii Kirk

Fountain Emily D. 1 3 efountain@gmail.com
Malumbres-Olarte Jagoba 2 3
Cruickshank Robert H. 3
Paterson Adrian M. 3
1 Department of Forest and Wildlife Ecology, University of Wisconsin-Madison , Madison, WI , USA
2 Center for Macroecology, Evolution and Climate, Natural History Museum of Denmark, University of Copenhagen , Copenhagen , Denmark
3 Department of Ecology, Faculty of Agriculture and Life Sciences, Lincoln University , Lincoln, Christchurch , New Zealand
Huber Dezene
Electronic publication date: 2015 Feb 5
Publication date: 2015
Volume: 3
Electronic Location ID: e749
Received 2014 Nov 19; Accepted 2015 Jan 14
Copyright: © 2015 Fountain et al.
Copyright year: 2015
Copyright holder: Fountain et al.
License: This is an open access article distributed under the terms of the Creative Commons Attribution License, which permits unrestricted use, distribution, reproduction and adaptation in any medium and for any purpose provided that it is properly attributed. For attribution, the original author(s), title, publication source (PeerJ) and either DOI or URL of the article must be cited.
License URL: https://creativecommons.org/licenses/by/4.0/

Keywords: Population dynamics, Survey, Endemic, Genetic variability, Chatham Islands

Funding: Miss E L Hellaby Indigenous Grasslands Research Trust Lincoln University Research Fund Funding and support for this project was provided by the Miss E L Hellaby Indigenous Grasslands Research Trust and the Lincoln University Research Fund. The funders had no role in study design, data collection and analysis, decision to publish, or preparation of the manuscript.

==============================
Human alteration of islands has made restoration a key part of conservation management. As islands are restored to their original state, species interactions change and some populations may be impacted. In this study we examine the coxella weevil, (Hadramphus spinipennis Broun) and its host-plant Dieffenbach’s speargrass (Aciphylla dieffenbachii Kirk), which are both open habitat specialists with populations on Mangere and Rangatira Islands, Chathams, New Zealand. Both of these islands were heavily impacted by the introduction of livestock; the majority of the forest was removed and the weevil populations declined due to the palatability of their host-plant to livestock. An intensive reforestation program was established on both islands over 50 years ago but the potential impacts of this restoration project on the already endangered H. spinipennis are poorly understood. We combined genetic and population data from 1995 and 2010–2011 to determine the health and status of these species on both islands. There was some genetic variation between the weevil populations on each island but little variation within the species as a whole. The interactions between the weevil and its host-plant populations appear to remain intact on Mangere, despite forest regeneration. A decline in weevils and host-plant on Rangatira does not appear to be caused by canopy regrowth. We recommend that (1) these populations be monitored for ongoing effects of long-term reforestation, (2) the cause of the decline on Rangatira be investigated, and (3) the two populations of weevils be conserved as separate evolutionarily significant units.

Introduction

Island restoration is a key focus of conservation biology. Often islands are home to endemic flora and fauna which may be heavily impacted by human modification and introduced species (Hutton, Parkes & Sinclair, 2007). Although restoring an island to a pre-human state is ideal, its restoration is not straightforward (Lawton, 1997) given a lack of information on species interactions and pre-human community composition and ecosystem conditions. Islands are unlikely to ever be fully restored to their previous ‘pristine’ states (Atkinson, 1990). The dynamic nature of colonization from the mainland (Sinclair & Byrom, 2006) and a relative lack of literature regarding long-term management also hinder island restoration (Simberloff, 1990).

Despite these constraints, successful restoration has been achieved on small scales (Simberloff, 1990). Extensive ecological research is necessary for restoration efforts to be successful, and also competent genetic management of island species is pivotal for long-term success (Jamieson, Wallis & Briskie, 2006). Genetics plays an important role in ecology and conservation (Frankham, Ballou & Briscoe, 2002). The loss of genetic variation, common among island populations with low effective population sizes (Ne), can decrease fitness and lower ability to adapt to changing environments (Pertoldi, Bijlsma & Loeschcke, 2007). Island restoration often involves the translocation or captive rearing of organisms that are already genetically depauperate (Jamieson, Wallis & Briskie, 2006). Understanding of the genetic characteristics of species involved in island restorations is vital for long-term viability of the species, and the success of a restoration project.

Islands within the New Zealand archipelago have been actively restored for many years, using pest eradication, replanting, translocation, captive rearing and reintroductions (Towns et al., 2012). One group of such islands is the Chatham Islands (Chathams), which is approximately 800 km east of New Zealand’s South Island and consists of two large populated islands (Chatham and Pitt) and a series of smaller, uninhabited islands. The Chathams have a history of intense geological activity that resulted in frequent submerging and re-emerging of the islands (Heenan et al., 2010). The most recent emergence of the smaller islands, such as Mangere, is thought to have been between 3.0 and 2.0 Ma (Heenan et al., 2010). Two of the Chatham islands, Rangatira and Mangere, have been under restoration since 1961 and 1968, respectively. The two islands are home to several rare birds [e.g., Petroica traverse (Chatham Island black robin], plants [e.g., Myosotidium hortensia (Chatham Island forget-me-not)] and invertebrates [e.g. Amychus spp. (Chatham’s giant click beetle)].

The coxella weevil (Hadramphus spinipennis Broun 1911) is a large, flightless weevil endemic to the Chatham Islands. There are four species in the genus Hadramphus (Craw, 1999), all endemic to New Zealand, three of which are listed as threatened or vulnerable, including H. spinipennis (Hitchmough et al., 2007). The extant populations of this species are found on Rangatira, Little Mangere and Mangere Islands, but the weevil was also historically recorded on Pitt Island in 1900 (Emberson et al., 1996). Surveys of Mangere and Rangatira Islands (Schöps, 1998) suggested that the weevil was thriving with an estimate of over 10,000 weevils found in the 1995/1996 summer on Mangere Island (Schöps, 2002). The weevil is associated with the plant Dieffenbach’s speargrass (Aciphylla dieffenbachii), which is found in open habitats on the islands. However, it has also been sighted on Pseudopanax chathamicum Kirk (Araliacene) several hundred meters away from A. dieffenbachii (A Liddy & G Taylor, pers. comm., 2014; Emberson et al., 1996). Choice tests between A. dieffenbachii and P. chathamicum suggest that Pseudopanax is most likely not a viable host plant for this weevil (Schöps, Wratten & Emberson, 1999). Hadramphus spinipennis is often found during September to February, feeding and mating on its host plant on warm, humid nights (Schöps, Wratten & Emberson, 1999).

Schöps (2000) performed behavioral and ecological studies on H. spinipennis and its host plant, A. dieffenbachii, on Mangere Island between 1993 and 1997, and on Rangatira Island in the summer of 1995/6. The main driver for this study was the observation of local extinctions of the host-plant on Mangere, which were thought to be caused by the weevil. Local extinctions of the host-plant was caused by the weevils before relocating to a new patch of A. dieffenbachii (Schöps, 1998). Schöps (2000) recommended, as part of the conservation management strategy (Department of Conservation, 1999), that H. spinipennis and A. dieffenbachii should be monitored every three to four years on Mangere and Rangatira to ensure that the regenerating forests do not affect the population dynamics and survival of the weevil and its host-plant.

Our study was conducted 13 years after the initial research by Schöps (2000) and has two main aims: (1) to measure the genetic similarity between the two populations and the genetic diversity within the species as a whole, and (2) determine if the current population dynamics are consistent with those found by Schöps and whether they support her original prediction of stable metapopulation dynamics. We hypothesize that (1) geographic isolation inhibits gene flow between weevil populations and (2) loss of open-habitat has caused a decline in the A. dieffenbachii populations defined by Schöps over a decade ago. We performed DNA analysis on weevils from both islands and repeated the surveys conducted on Mangere and Rangatira Island. Our study explores the possible negative impacts of restoration on an endangered invertebrate by incorporating a genetic and ecological approach.

Methods

Site descriptions

Rangatira (South East) Island is 219 ha and was heavily farmed until its purchase by the New Zealand government in 1953, after which all livestock were removed in 1961 (Department of Conservation, 2012) (Fig. 1). Much of the original forest was destroyed for farming and many of the native plants that remained were grazed by livestock. Currently, the island is mostly covered by remnant or regenerating forest and A. dieffenbachii is limited to the coastal cliffs and rocky shores.

Figure 1 Map of Chatham Islands.

Map (1:500,000 scale) of the Chatham Islands with Rangatira (South East) and Mangere Islands labeled. Insert map shows the location of the Chatham Islands in relation to New Zealand.

Mangere Island is 113 ha and surrounded by cliffs, with the highest cliff reaching 286 m (Fig. 1). The island was once covered with native forest but 90% of the forest was burned for sheep farming and many of the native vascular plants and megaherbs were suppressed by heavy grazing (Ritchie, 1970). Several plants, such as A. dieffenbachii, started to regenerate with the removal of livestock in 1968 and an intensive forest planting program was implemented in 1974 (Butler & Merton, 1992). Aciphylla dieffenbachii grows along the steep, rocky cliffs of Mangere and in the open grasslands. The plant has a patchy distribution over the whole island.

Survey of Hadramphus spinipennis and Aciphylla dieffenbachii

Rangatira (South East) Island (14–19 February 2010) and Mangere Island (17–23 February 2011) were surveyed for A. dieffenbachii during the day. The search was restricted to coastal and open areas where A. dieffenbachii has been documented to grow (Schöps, 2000). Due to adverse weather conditions, the high cliffs of both islands could not be surveyed although on Rangatira, although there had been reports of a large A. dieffenbachii population on North Summit.

When plants were found, the quantity and state of flowering and a visual assessment of plant size were made. Plants were searched for signs of weevil feeding and activity, in particular the characteristic margin feeding of H. spinipennis (Schöps, 2000). Although other herbivores can be found on A. dieffenbachii, the feeding pattern of H. spinipennis is distinctly different and well documented by photograph, making feeding signs easily observable (Fig. 2). Each plant was photographed and the surrounding area was searched for new seedlings. GPS coordinates of major plant clusters were recorded (Table S1). At night, starting at 22:00, known A. dieffenbachii populations were visually searched for H. spinipennis individuals. Data collection on Mangere was conducted in the same manner as on Rangatira. All locations where A. dieffenbachia and H. spinipennis had previously been recorded were visited and surveyed.

Figure 2 Weevil marginal feeding pattern.

A photograph depicting the weevil feeding pattern, which is identified by marginal notches. Photograph by: Jagoba Malumbres-Olarte.

DNA collection and PCR

Due to collection restrictions of protected species by the Department of Conservation and also local imi/iwi, weevils for DNA analysis were limited to 15 individuals captured per island. Adult H. spinipennis range from 18–22 mm in length. Individuals were randomly selected from different subpopulations on the islands; the tarsal claw and the first two segments of the tarsus were removed using ethanol-cleaned scissors and the weevil was then released. Previous work has shown that a tarsal clip has no known negative impacts on the weevils (Fountain et al., 2013). Clips were stored in propylene glycol and, when returned to the laboratory, were washed with 95% ethanol and then stored at −20 °C in 95% ethanol. Each tarsal clip was cut into several pieces using a sterile scalpel blade and then transferred into a 1.7 ml Eppendorf tube. A QIAmp DNA Investigator Kit (Qiagen, Auckland, catalog # 56504) was used for DNA extraction following the manufacturer’s protocol for tissue samples.

Two mitochondrial genes, cytochrome c oxidase subunit I (COI) and cytochrome b (cytb), and one nuclear gene, internal transcribed spacer 2 (ITS2), were amplified by polymerase chain reaction (PCR). For all PCRs, 2.5 µl of the DNA extraction was added to the following: 2.5 µl of 0.25 mM of dNTPs, 0.2 µl of polymerase, 1 µl of 20 µM for each primer, 2.5 µl of 10× PCR buffer (i-taq; iNtRON Biotechnologies) and deionized water to bring the total reaction volume to 25 µl.

COI was amplified using the primers LCO1490 and HCO2198 (656 base pair fragment) (Folmer et al., 1994). The PCR cycle was 94 °C for 3 min followed by 35 cycles of 94 °C for 45 s, 45 °C for 45 s and 72 °C for 1 min 20 s, with a final extension at 72 °C for 5 min. The primers CB1 and CB2 (432 base pair fragment) (Simon et al., 1994) were used to amplify cytb. The PCR cycle was 94 °C for 3 min followed by 40 cycles of 94 °C for 30 s, 49 °C for 45 s and 72 °C for 1 min, with a final extension at 72 °C for 5 min.

Due to the difficulty in amplifying a large fragment of ITS2 using the original primer set, a new genus-specific primer set was developed for a 450 base pair fragment of ITS2 (Table 1) (see Appendix S1 for primer design details). The PCR mixture and cycle for these primers were the same as those for ITS3 and ITS4, except that the annealing temperature was decreased to 54 °C. Every PCR reaction included a negative (water) control with no DNA.

Table 1 ITS2 primers.

Primers developed to amplify a region of the ITS2 gene in H. spinipennis.

Primer name	Primer sequence 5′ to 3′	
Had ITS2 For	ATT CTG TTC CCG GAC CAC TCC TGG CTG A	
Had ITS2 Rev	GCG CGC ACC GTT ACR ATC KGA CGY C	

Molecular data analysis

Sequence chromatograms for 30 COI, 16 cytb and 25 ITS2 sequences were visualized using FinchTV 1.4 (Geospiza) and forward and reverse sequences were manually aligned in Mega 5.05 (Tamura et al., 2011). No insertions, deletions or stop codons were found for COI or cytb. Within ITS2 a variable AT short tandem repeat (STR) was found in the middle of the sequences. Short tandem repeats may have higher mutation rates compared to the flanking regions which may interfere with the phylogenetic signal (Selkoe & Toonen, 2006) so we conducted preliminary analysis on the ITS2 data set with and without the STR. Preliminary phylogenies were constructed in MEGA from a Kimura two-parameter (K2P) distance matrix (Kimura, 1980) using neighbor joining (NJ) with 1,000 bootstrap replicates. No differences were found in the ITS2 analysis when the STR was removed; we chose to use the dataset without the STR for additional analysis to be confident the STR would not interfere with the phylogenetic inference.

For Bayesian analysis, the COI data set was reduced to 16 sequences to match them with the cytb sequences. Sequences of the two genes were then concatenated in R 2.13.2. (R Development Core Team, 2011). The concatenated mitochondrial sequence data were analyzed separately from the ITS2 data because they are independently evolving loci and also have differences in mutation rates. The ITS2 analysis was performed on the full set of 25 sequences. Bayesian analyses were performed using Beast 1.7.1 (Drummond et al., 2012). The best partitioning scheme and evolutionary model were found with PartitionFinder (Lanfear et al., 2012) using the corrected Akaike Information Criterion (AICc) for COI and cytb. The Kimura three-parameter (K81) model (Kimura, 1981) was chosen with no partitioning between the concatenated genes (known collectively below as mtDNA). Since ITS2 is not a protein coding gene it was not considered for partitioning, and the best fit evolutionary model was found using the AICc with jModeltest 2.1.1 (Darriba et al., 2012), which identified the symmetrical (SYM + G) model (Zharkikh, 1994) with gamma distribution as the optimal model.

The mtDNA and ITS2 data were analyzed under a strict molecular clock with a coalescent (constant population size) tree prior. For each gene, four replicate runs of a chain run for 50 million generations were performed, sampling every 2000 generations. Convergence and effective sample size of each parameter was assessed in Tracer 1.5 and samples from the four runs were pooled using Log Combiner 1.7.2. After discarding the initial 10% as burn-in, a maximum clade credibility tree was compiled in TreeAnnotator 1.7.1 (Drummond et al., 2012).

For the mitochondrial data, two Bayesian analyses were conducted to provide estimates of divergence time using (i) the geological evidence and the timing of the volcanic emergence of Mangere, 3–2 million years ago (Heenan et al., 2010), and (ii) the standard invertebrate mitochondrial rate of 0.0115 substitutions/site/million years (Brower, 1994). For geological dating, priors were set to allow tree calibration using direct input of a fixed date for specific nodes; the root of the tree was constrained to no older than 3 million years with uniform prior. Since we aim to estimate the split between the Rangatira and Mangere populations, a distribution of dates was not used as Mangere did not emerge until 3 million years ago at the earliest. All analyses were performed in BEAST 1.7.2 (Drummond et al., 2012) using the same model and partitioning scheme employed in the previous analysis. Four independent runs consisting of a chain run for 50 million generations were conducted, sampling every 2,000 generations. Maximum clade credibility trees were generated in the same manner as in the previous analysis.

Pairwise genetic distances for mtDNA were taken from the Bayesian maximum clade credibility tree using PASSaGE 2.0 (Rosenberg & Anderson, 2011). A Neighbor-Net network of ITS2 haplotypes was generated using SplitsTrees 4.12.3 (Huson & Bryant, 2006) to visualize conflicting patterns in the phylogenetic signal. Distances for the Neighbor-Net network were calculated under the generalized time reversible (GTR) model and no rate heterogeneity among sites.

Results

Survey

Rangatira

Three locations were found to have subpopulations of A. dieffenbachii: East Clears, West Clears, and West Landing (subpopulations A, B, C in Fig. 3). Subpopulation A showed signs of heavy weevil herbivory on all plants. In subpopulation B only a few plants showed signs of weevil damage, whereas subpopulation C had no weevil feeding damage. Table 3 provides a detailed description of the number of plants, size of plants, and flowering state for the A. dieffenbachii populations.

Figure 3 Maps of Rangatira survey sites.

Rangatira Island with the populations of A. dieffenbachii that were recorded in the 1995 survey by Schöps (2000) (white circles) and the A. dieffenbachii populations found in the 2010 survey (white circles with letters). Letters were assigned to each site as they were found. The triangle with a question mark on the 2010 map represents the one area that could not be surveyed due to adverse weather conditions. Map image was obtained from Google Earth 2013 image and the same image was used for both maps. Map data: Data SIO, NOAA, U.S. Navy, NGA, GEBCO. Image from Google Earth and ©2013 Digitalglobe.

In subpopulation A, 29 H. spinipennis were found feeding and mating on the plants. The male: female sex ratio of the weevils was 15:14. In subpopulation B, three out of the nine major A. dieffenbachii groups had weevils. Group 1 had two males and two females: one pair breeding and the other two feeding. Group 4 had one breeding pair, and group 9 had one male feeding.

West Landing was searched for two nights but no weevils were found (Fig. 3). One male H. spinipennis was found on Pseudopanax chathamicus, approximately 250 m from the nearest A. dieffenbachii at West Landing.

Mangere

Seven subpopulations of A. dieffenbachii were identified on Mangere (Fig. 4). Feeding signs were observed in plants from all subpopulations. In addition to the plants found in the seven subpopulations, a few individual plants were also found scattered along the south-east coast of the island but had no evidence of weevil feeding damage (Fig. 4). A detailed description of the A. dieffenbachii populations, including number of plants, size of plants and flowering state, is provided in Table 4.

Figure 4 Map of Mangere survey sites.

Mangere Island with the A. dieffenbachii sites that were recorded in the 1995 survey by Schöps (2000) (white areas) and the A. dieffenbachii populations found in the 2011 survey (white areas with numbers representing the larger patches of A. dieffenbachii). Numbers were assigned to each site as they were found. The entire island was surveyed visually for A. dieffenbachii, but some plant patches could not be reached for hand-searching for weevils. Map image was obtained from Google Earth 2013 image and the same image was used for both maps. Map data: Data SIO, NOAA, U.S. Navy, NGA, GEBCO. Image from Google Earth and ©2013 CNES / Astrium.

Individuals of H. spinipennis were found in five of the seven A. dieffenbachii subpopulations. The observed number of weevils reached 26 in location 1, which corresponded to subpopulation 1 of A. dieffenbachii. Ten specimens were counted in locations 4, 6 and 3, and only two in location 2. The male: female sex ratio ranged from 19:7 in location 1, through 8:6 in location 4, to 1:1 in locations 3 and 2. Weevils were observed feeding and mating in all locations except for location 2, where the two individuals were feeding on separate plants.

Molecular analysis

In total, 30 specimens of H. spinipennis were successfully sequenced for COI, 16 for cytb and 25 for ITS2. For COI, a 656 bp fragment was obtained, which had three variable sites, two of which were parsimony informative. Base frequency means for COI were unequal and AT-rich (T = 35.0%, C = 19.6%, A = 30.3% and G = 15.1%); a chi-square test confirmed heterogeneity of base frequencies across all taxa (d.f. = 87, p = 1.00). For cytb, a 432 bp fragment was obtained, which had one variable site, which was parsimony informative. Base frequency means for cytb were unequal and AT-rich (T = 36.6%, C = 22.2%, A = 29.3%, and G = 11.9%); a chi-square test confirmed heterogeneity of base frequencies across all taxa (d.f. = 45, p = 1.00). The 30 COI sequences showed little variation with only two individuals sharing a single nucleotide difference between themselves and the rest of the samples. However, for cytb there was a clear difference at one nucleotide in all samples between individuals from each of the two islands. The pairwise genetic distances in the concatenated mitochondrial genes were low, with a mean of 0.002 (max 0.003) substitutions/site between the two island populations. The maximum P-distance within each island was 0.001 substitutions/site.

After the removal of the AT variable repeat, a 428 bp fragment was obtained for ITS2 which contained 24 variable sites, 17 of which were parsimony informative. Four indels were found in the sequences: two from an individual on Rangatira and two from an individual on Mangere. The Neighbor-Net network does not show strong genetic structure within or between the islands (Fig. 5). The results of the Neighbor-Net network (Fig. 5) suggest that there may be a complicated signal between the two islands represented by the network separating the two groups.

Figure 5 ITS2 Neighbor-Net network.

Neighbor-Net network generated from ITS2 distances. The underlined numbers are individuals from Mangere and the letters are individuals from Rangatira.

The maximum clade credibility trees were very similar in topology to the NJ trees; due to the splitting of zero-length branches in the maximum clade credibility trees we opted to display the NJ trees. The mitochondrial tree shows a weakly-supported split between the two islands (Fig. 6A), and the branch length separating the two islands is extremely small at 0.002 substitutions/site. The universal mitochondrial rate for invertebrates of 0.0115 substitutions/site/million years places the origin of this split around 1,300 BP (95% HPD 0.000–0.0139 BP) and the geological time calibration of 4 million years places it around 600 BP (95% HPD 0.0001–0.0016 BP) (Fig. 6A and Table 2). The ITS2 tree shows a lack of resolution and does not support any split between the two islands (Fig. 6B). There is no evidence for genetic differentiation in COI and ITS2 among the subpopulations for each island [subpopulations A, B, C for Rangatira and subpopulations 1–7 for Mangere (Figs. 3 and 4)].

Figure 6 Neighbor joining trees for mitochondrial genes and ITS2.

(A) Neighbor joining (NJ) tree for the concatenated mitochondrial genes COI and cytb. (B) Neighbor joining tree for ITS2. For both trees, individuals from Mangere are underlined (first number of weevil ID is subpopulation and the second number is ID to denote different weevils). The maximum clade credibility tree posterior probabilities higher than 95% are labeled below the branch node and the bootstrap value for the NJ tree are above the branch node. The root that was estimated in Beast for the maximum clade credibility tree and it is represented by the grey root line. The scale bar is in substitutions/site.

Table 2 Divergence times for the mitochondrial maximum clade credibility tree.

Mean divergence times (Ma) and 95% confidence interval for the two different dating schemes used to date nodes in the mitochondrial maximum clade credibility tree. Dates were estimated in BEAST using a strict molecular clock with fixed mean rate of 0.0115 substitutions/site/my, and a strict molecular clock with a 3 million year age constraint on the tree root.

	Divergence times (Ma)	
	Fixed mean rate	Geological date	
Mean	0.0013	0.0006	
95% confidence interval	0.0000–0.0139	0.0001–0.0016	

Table 3 Aciphylla dieffenbachii populations on Rangatira.

Aciphylla dieffenbachii populations found on Rangatira including estimated population size, approximate size of the plants found in the population and whether the plants were in flower.

Population	A	B	C	
Number of plants	10	3 to 6a	2 to 7 and 55a	
Plant size	medium	Small/medium	Small to large/seedlings	
Flowering	No	No	Yesb	
Notes.

a Population B had nine groups of plants spread across a cliff face; each group consisted of 3 to 6 plants. Population C had five groups of plants patchily distributed with 2 to 7 plants in each group. A total of 55 seedlings were spread throughout the area.

b Only two female plants were flowering.

Table 4 Aciphylla dieffenbachii populations on Mangere.

Aciphylla dieffenbachii populations found on Mangere including estimated population size, approximate size of the plants found in the population and whether the plants were in flower.

Population	1	2	3	4	5	6	7	
Number of plants	410	100	180	330	20	150	17	
Plant size	Small/medium	Small to large	Small to large	Small/medium	Small	Small/medium	Small	
Flowering	Yes	Yes	Yes	Yes	Yes	Yes	No	

Discussion

The genetic analysis shows a population of weevils that is highly similar across the islands with only a small amount of difference between the two islands in the mitochondrial genes. A genetic difference between the weevil populations of Mangere and Rangatira has been previously reported for a different section of the COI gene (Goldberg & Trewick, 2011). The analysis with the two concatenated mitochondrial genes showed a strong support for a split between the two islands. This difference is not seen in the nuclear gene ITS2; however, due to the longer coalescent times of nuclear genes compared to mitochondrial genes, there may have been insufficient time for ITS2 to achieve reciprocal monophyly. Although some signal interference in the ITS2 gene caused by incomplete lineage sorting and recent divergence can skew the resulting phylogeny, there is still strong support for a group that includes two individuals, one from each island. Further molecular work should be conducted on more nuclear genes to discover a clearer signal of genetic differentiation between the two islands.

The difference in the mitochondrial and nuclear gene trees can be explained by the four times faster coalescence time in mitochondrial genes compared to nuclear genes (Ballard & Whitlock, 2004). The island split in the mitochondrial gene tree may be the result of the two populations having no gene flow. Given the faster mutation rate of mitochondrial genes versus the nuclear genes (Moriyama & Powell, 1997), reproductive isolation is likely to lead to faster differentiation of the mitochondrial genes. In the future, this split may become evident in the slower evolving nuclear genes if the populations remain genetically isolated. Currently, the populations of the weevil and its host plant on Rangatira are not as heavily monitored as they are on Mangere. If the Rangatira and Mangere populations are confirmed to be genetically isolated, it is of utmost importance to take conservation measures to make sure that both populations have similar chances of survival.

Both methods of dating the separation between individuals on the two islands using the mitochondrial genes date the mean time for this split as rather recent, in the last few thousand years. This timing approximately coincides with the arrival of humans and introduced mammals, particularly Polynesian rats, which are known to predate Hadramphus weevils (Towns, 2009). Calibrating the molecular clock using the geological age of 4 million years as the last emergence of the islands gave a smaller 95% confidence interval. Given that using a universal mitochondrial rate is fraught with problems, such as variation between genes, differences depending on substitution model used and the universal rate does not account for variance in coalescent times (Papadopoulou, Anastasiou & Vogler, 2010), geological dating can offer a better estimation of the time of separation between the two islands when applying the correct distribution to account for dating uncertainty (Ho, 2007).

The lack of genetic diversity found within H. spinipennis may be a result of population decline in the weevils. A likelihood of population decline is supported by the extinction of populations of H. spinipennis from Pitt Island in the late 1800s due to the loss of habitat and habitat modification from humans. Loss in genetic variation and extinction of populations points directly to a population bottleneck in H. spinipennis.

The results of this study suggest that there has been no decline in the number of the H. spinipennis and A. dieffenbachii populations surveyed on Mangere and population numbers have not been affected by the forest regeneration. The weevil has a consumer-resource metapopulation relationship with its host plant and will decimate populations of A. dieffenbachii before moving onto another population of plants (Schöps, 1998). Metapopulation theory suggests that as long as recolonization by the consumer exceeds or equals the extinction rate of the resource, the metapopulation will persist (Taylor, 1990). On Mangere, there is evidence for subpopulations going extinct while new A. dieffenbachii populations have arisen, which suggests a dynamic system of localized host-plant exploitation, extinction, and weevil dispersal.

On Rangatira, there has been a decline in both H. spinipennis and A. dieffenbachii populations. Although two of the populations (A & B) of A. dieffenbachii were found in 1995 and 2010, the plants were in decline in 2010 with no flowering and no large-sized plants. One population of A. dieffenbachii that was found in 1995 was no longer in existence, not even as seedlings, in 2010. The newly discovered population of plants on the West Clears contained mainly small plants and seedlings, with only one plant that flowered, and may not be able to support a large weevil population. The decline on Rangatira does not seem to be the result of forest regeneration, as the locations of all A. dieffenbachii populations are on open grasslands and cliffs where no forest has regenerated. Modeling suggests that if the distance between host-plant populations is great enough to not be easily traversed, but not so far that it is unreachable by the weevils, then the consumer-resource dynamic could persist for the long-term (Johst & Schöps, 2003). The host-plant populations on Rangatira were separated by several hundred meters and the weevil is known to travel up to 500 m to a new plant resource (Schöps, 1998). Therefore, distance between populations does not appear to be the cause of its decline. More intensive monitoring should be conducted on Rangatira to determine the cause of the host-plant decline and whether the metapopulation dynamic between H. spinipennis and A. dieffenbachii has been altered.

Hadramphus spinipennis is listed as endangered by the New Zealand Department of Conservation and is considered a species of concern due to its very restricted range. Although spatial models have shown that the weevil population is in equilibrium, and could possibly be classified as “common” (Kean, 2006), combining genetics with longer-term ecological data does not completely support the notion that H. spinipennis is common on both Mangere and Rangatira. In regards to the number of weevils and host-plant populations on Mangere, the weevil seems relatively abundant; however, on Rangatira lower weevil numbers were found. Genetic data indicates that both populations do not possess a large amount of genetic diversity and have most likely undergone a population bottleneck. Population bottlenecks are related to a loss of genetic diversity which can have significant consequences for long-term viability of small populations (Grueber, Wallis & Jamieson, 2008). As the forests on Mangere and Rangatira Islands continue to regenerate, A. dieffenbachii may decline due to a loss of open habitat. Although the population counts suggest no change in H. spinipennis on Mangere, the weevil may be unable to adapt to a possible decline in its host-plant resulting from the loss of open habitat. This is particularly a concern on Rangatira where a decline in weevil numbers was seen.

Conclusions

A pattern of genetic structure separating a species by island haplotype is something that has been reported in other flightless beetles (Sequeira et al., 2012; Stroscio et al., 2011). Between the two islands there are some genetic differences which should be preserved if possible, as the H. spinipennis on Rangatira and Mangere Islands are shown to be genetically isolated from each other. The unique haplotypes from each island adds variation and allows for the possibility of cross-introduction of weevils to preserve genetic diversity. Our results strongly suggest that forest restoration on Rangatira and Mangere Islands has not had a negative impact on H. spinipennis or its host plant; however, a long-term survey, in particular accessing the area on Rangatira that could not be searched in this study, is recommended to confirm small population sizes. Although continued monitoring of the weevil populations is recommended, forest restoration should continue and is not impacting on the conservation of an open habitat specialist.

Supplemental Information

Appendix S1 ITS2 primer design

Click here for additional data file.

Supplemental Information 2 ITS2 alignment

The ITS2 alignment for Mangere and Rangatira; the STR is included in this alignment.

Click here for additional data file.

Table S1 GPS coordinates of survey sites

GPS coordinates in degrees, decimal minutes of the Aciphylla dieffenbachii subpopulations surveyed on Rangatira and Mangere Islands.

Click here for additional data file.

Table S2 GenBank accession numbers

Click here for additional data file.

We thank the Chatham Islands Conservation Board, the Chatham Island imi/iwi, and the Department of Conservation (DoC) for the permit (WE-26391-RES) to collect and for access to the island, housing, and general assistance.

Additional Information and Declarations

Competing Interests

Author Contributions

Field Study Permissions

DNA Deposition

The authors declare there are no competing interests.

Emily D. Fountain conceived and designed the experiments, performed the experiments, analyzed the data, wrote the paper, prepared figures and/or tables.

Jagoba Malumbres-Olarte performed the experiments, analyzed the data, wrote the paper, reviewed drafts of the paper.

Robert H. Cruickshank and Adrian M. Paterson contributed reagents/materials/analysis tools, reviewed drafts of the paper, supervisory team.

The following information was supplied relating to field study approvals (i.e., approving body and any reference numbers):

Department of Conservation (DoC) granted permission for the permit WE-26391-RES.

The following information was supplied regarding the deposition of DNA sequences:

The sequences will be uploaded to GenBank and accession numbers are provided in the Supplemental Information.

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
