# Peer review of "The effects of island forest restoration on open habitat specialists: the endangered weevil Hadramphus spinipennis Broun and its host-plant Aciphylla dieffenbachii Kirk"

_PeerJ, doi:10.7717/peerj.749_

## Round 0.1 · original submission · Minor Revisions

Thank you for your submission to PeerJ. This is a well-written and interesting paper that covers an important conservation topic and opens space for new research on these weevils. Both reviewers have suggested minor revisions, and I concur. Both reviewers have given a number of important recommendations which the authors should consider and respond to. The authors should be sure to notice that one reviewer placed many substantive comments on the PDF file attached to their review; there are also comments associated with the figures in that review.

A few extra considerations:

Line 57: use square brackets as outside parentheses when you have a nested set like this.

Line 66: Capitalize the genus, and italicize the binomial, for Coxella dieffenbachii.

In general I note that Aciphylla dieffenbachii and Coxella dieffenbachii seem and to be taxonomic synonyms, with the former in current use. The general use of Coxella dieffenbachii in the abstract and then the introduction with differences in capitalization and italics makes things a bit confusing, though. Perhaps is it better to just stick with Aciphylla dieffenbachii and Dieffenbach’s speargrass (i.e., dispense with Coxella dieffenbachii) throughout since this seems to be the currently accepted binomial and common name?

Please include GPS coordinates for survey sites and for the various collection sites of the 2 x 15 weevils (the methods note that these were, appropriately, recorded). This information would likely be best presented in a table or perhaps two tables (one for survey sites, and one for weevil collection sites). While the map locations are reasonable, they are still too large to guide future researchers or conservationists to the exact areas.

Line 152 should read either: “…analysis was performed on the full set of 25 sequences.” or “…analysis was performed on a full set of 25 sequences.”

Line 285: should be “utmost”, not “upmost”

Section around line 285: After responding to reviewer’s comment about reproductive isolation, you will likely have to change the wording, etc., here. Depending, of course, on your response/rebuttal.

Recent work in our research program involves monitoring the host-finding movements of Hylobius warreni, a flightless weevil that uses pine hosts. In some respects (size, flightlessness, etc.) H. spinipennis and H. warreni seem quite similar. Anecdotal evidence from our work suggests that climatic conditions can lead to either reduction in movement or fairly rapid death in what is otherwise a rather robust organism. Specifically in the case of H. warreni, hot dry weather causes the insects to either burrow underground (unless they are already under a host tree) or die. While I obviously do not know the landscape on your study islands, having never been there, is it possible that habitat degradation has changed the canopy structure or other structural aspects such that weevil population declines are due to mortality or lack of mate-finding/mating during particular climatic events; particularly extended events such as overly dry, hot spells?

Please consider making the review history of this MS public, as it is an important and useful part of the scientific record.

·

Basic reporting

Great. No comments.

Experimental design

See general comments. Overall, good design, but I did raise some questions about sample size.

Validity of the findings

Good -although generality can be questioned - see general comments (below)

Additional comments

Review of weevil paper by Fountain et al.

The paper by Fountain et al. is quite interesting; it presents a terrific system to study metapopulations, conservation, evolution & population genetics. The study species itself is also ideal- just "common" enough to do high quality science, but obviously rare enough to warrant significant conservation concern. The links to land use, history, and to the plant communities are fascinating and certainly sets the stage well.

Overall I found the paper to be well written, clear, with a good design, analyses and presentation of results. The authors are to be commended: it's a real treat to read a paper that is well packaged and presented.

This is a solid piece of science, and I have relatively few specific criticisms. Although the geographic scope of the work is narrow, the author's approach of combining population genetics with field surveys Is excellent, and will be of interest to many. Also, the conservation message is important, and relevant to researchers and conservationists around the world.

Below are a few comments that I believe the authors should take under consideration when they work on revising the paper.

Parts of the introduction are a little misleading: it sets the stage for a project that will indeed look at multiple metrics for restoration success (eg, diversity, vegetation, ecological process), yet the data are mostly about the weevil, and the bulk of the story rests with genetic diversity. I think either the vegetation data needs to be bolstered considerably (eg, formally mapped spatially, with beetle data overlaid?), or the intro should be packaged and framed more around genetic diversity and conservation, viable population sizes, etc. The framing around restoration and conservation is good, but the data just don't match that framing as strongly as perhaps they could or should. So, I think the overarching framework needs some thought and perhaps a fairly significant overhaul.

Related, I think the introduction could be more concise, and perhaps 2/3 of its current length.

I wonder whether 15 individuals is a high enough sample size. I think some wording in the text will be needed to justify this sample size and reassure the readers that this is adequate. I'm afraid I don't know the literature on this topic well enough to know, but this question does come to mind!

It's not explicitly clear about where the tarsal clips come from ie, from what individuals? I assume they are from individuals collected as part of the survey rather than from museums, for example. This requires some clarifications, and I wonder if the methods ought to be reorganized so the reader first understands the survey part followed up by the genetic methods

Are there other herbivores on the plants? How confident are the authors in the observations of herbivory by the weevil? This needs to be clarified and some idea if the degree of confidence in the herbivory data needs to be explicitly stated.

Line 290 - it's not clear here if there is a potential relationship between the rats & the weevils, or that is just meant to be a general statement.
Line 310-311 : this is perhaps too speculative - this would require a very different kind of study, ie, on dispersal events, etc

Perhaps my most significant concern is that the conservation implications may also be overstated - the data are strong and the genetic diversity results strongly support the main conclusion about differentiation among the islands, but I worry that the survey results aren't in-depth enough and perhaps additional effort with surveys (including on those hard-to-reach cliff faces) is needed to make the case. It's so hard to prove "small population numbers" without very extensive and long-term surveys, and although strongly suggestive, some parts of the conclusions need to be toned down.

Reviewer 2 ·

Basic reporting

Writing is generally very good. The attached pdf have notes on issues that need some improvements.

Experimental design

Please make the 'aims' more hypothesis driven. The paper sounds more like a survey, but in fact you are testing several hypotheses, e.g. there is no gene flow between populations (unique populations); genetic variability in small vulnerable populations is depauperate; habitat size is decreasing.

Some results are well known such as AT-bias etc and should be removed from the results section. More details is needed in the ITS variation.

Please provide an alignment for the ITS data.

Consider if a NJ tree will improve readability; Beast produces trees with arbitrary splits between zero-length branches.

Splits tree is a good solution for the ITS data.

Validity of the findings

The authors can be more confident in their population divergence estimates, see pdf. Also, the most important feature of having unique haplotypes on different islands is the contribution to greater metapopulation variation and the possibility for cross-introductions of beetles.

Annotated reviews are not available for download in order to protect the identity of reviewers who chose to remain anonymous.

---

## Round 0.2 · accepted · Accept

The initial review resulted in two recommendations of minor revision, which I agreed with too. The authors have responded to all of the reviewers' and my comments and suggestions, and have done so adequately. This MS is now acceptable for publication in PeerJ.